# Head-to-Head Comparison of Aptamer- and Antibody-Based Proteomic Platforms in Human Cerebrospinal Fluid Samples from a Real-World Memory Clinic Cohort

**DOI:** 10.3390/ijms26010286

**Published:** 2024-12-31

**Authors:** Raquel Puerta, Amanda Cano, Pablo García-González, Fernando García-Gutiérrez, Maria Capdevila, Itziar de Rojas, Clàudia Olivé, Josep Blázquez-Folch, Oscar Sotolongo-Grau, Andrea Miguel, Laura Montrreal, Pamela Martino-Adami, Asif Khan, Adelina Orellana, Yun Ju Sung, Ruth Frikke-Schmidt, Natalie Marchant, Jean Charles Lambert, Maitée Rosende-Roca, Montserrat Alegret, Maria Victoria Fernández, Marta Marquié, Sergi Valero, Lluís Tárraga, Carlos Cruchaga, Alfredo Ramírez, Mercè Boada, Bart Smets, Alfredo Cabrera-Socorro, Agustín Ruiz

**Affiliations:** 1Ace Alzheimer Center Barcelona, Universitat Internacional de Catalunya, 08029 Barcelona, Spain; rpuerta@fundacioace.org (R.P.); acano@fundacioace.org (A.C.); pgarcia@fundacioace.org (P.G.-G.); fgarcia@fundacioace.org (F.G.-G.); mcapdevila@fundacioace.org (M.C.); iderojas@fundacioace.org (I.d.R.); colive@fundacioace.org (C.O.); jblazquez@fundacioace.org (J.B.-F.); osotolongo@fundacioace.org (O.S.-G.); amiguel@fundacioace.org (A.M.); lmontrreal@fundacioace.org (L.M.); aorellana@fundacioace.org (A.O.); mrosende@fundacioace.org (M.R.-R.); malegret@fundacioace.org (M.A.); vfernandez@fundacioace.org (M.V.F.); mmarquie@fundacioace.org (M.M.); svalero@fundacioace.org (S.V.); ltarraga@fundacioace.org (L.T.); mboada@fundacioace.org (M.B.); 2PhD Program in Biotecnology, Faculty of Pharmacy and Food Sciences, University of Barcelona, 08028 Barcelona, Spain; 3Biomedical Research Networking Centre in Neurodegenerative Diseases (CIBERNED), National Institute of Health Carlos III, 28029 Madrid, Spain; 4Departament de Farmacologia, Toxicologia i Química Terapèutica, Facultat de Farmàcia i Ciències de l’Alimentació, Universitat de Barcelona, 08007 Barcelona, Spain; 5Division of Neurogenetics and Molecular Psychiatry, Department of Psychiatry and Psychotherapy, Faculty of Medicine and University Hospital Cologne, University of Cologne, 50937 Cologne, Germany; pamela.martino-adami1@uk-koeln.de (P.M.-A.); alfredo.ramirez-zuniga@uk-koeln.de (A.R.); 6Janssen Pharmaceutica NV, a Johnson & Johnson Company, 2340 Beerse, Belgium; memon@its.jnj.com (A.K.); bsmets1@its.jnj.com (B.S.); acabrer8@its.jnj.com (A.C.-S.); 7NeuroGenomics and Informatics Center, Washington University School of Medicine, St. Louis, MO 63108, USA; yunju@wustl.edu (Y.J.S.); cruchagac@wustl.edu (C.C.); 8Hope Center for Neurological Disorders, Washington University, St. Louis, MO 63110, USA; 9Department of Clinical Biochemistry, Rigshospitalet, Copenhagen University Hospital, 2100 Copenhagen, Denmark; ruth.frikke-schmidt@regionh.dk; 10Department of Clinical Medicine, University of Copenhagen, 2200 Copenhagen, Denmark; 11Division of Psychiatry, University College London, London W1T 7NK, UK; n.marchant@ucl.ac.uk; 12Inserm, CHU Lille, Institut Pasteur de Lille, U1167-RID-AGE Facteurs de Risque et Déterminants Moléculaires des Maladies Liées au Vieillissement, Université de Lille, F-59000 Lille, France; jean-charles.lambert@pasteur-lille.fr; 13Institut Pasteur de Lille, Inserm U1167, CHU de Lille, LabEx DISTALZ, Université de Lille, F-59000 Lille, France; 14Department of Neurodegenerative Diseases and Geriatric Psychiatry, Medical Faculty, University Hospital Bonn, 53127 Bonn, Germany; 15German Center for Neurodegenerative Diseases (DZNE), 53127 Bonn, Germany; 16Department of Psychiatry and Glenn, Biggs Institute for Alzheimer’s and Neurodegenerative Diseases, San Antonio, TX 78229, USA; 17Cluster of Excellence Cellular Stress Responses in Aging-Associated Diseases (CECAD), University of Cologne, 50931 Cologne, Germany; 18Glenn Biggs Institute for Alzheimer’s & Neurodegenerative Diseases, University of Texas Health Science Center, San Antonio, TX 77204, USA

**Keywords:** proteomics, Olink, SomaScan, Alzheimer’s disease, mild cognitive impairment, cerebrospinal fluid, biomarkers

## Abstract

High-throughput proteomic platforms are crucial to identify novel Alzheimer’s disease (AD) biomarkers and pathways. In this study, we evaluated the reproducibility and reliability of aptamer-based (SomaScan^®^ 7k) and antibody-based (Olink^®^ Explore 3k) proteomic platforms in cerebrospinal fluid (CSF) samples from the Ace Alzheimer Center Barcelona real-world cohort. Intra- and inter-platform reproducibility were evaluated through correlations between two independent SomaScan^®^ assays analyzing the same samples, and between SomaScan^®^ and Olink^®^ results. Association analyses were performed between proteomic measures, CSF biological traits, sample demographics, and AD endophenotypes. Our 12-category metric of reproducibility combining correlation analyses identified 2428 highly reproducible SomaScan CSF measures, with over 600 proteins well reproduced on another proteomic platform. The association analyses among AD clinical phenotypes revealed that the significant associations mainly involved reproducible proteins. The validation of reproducibility in these novel proteomics platforms, measured using this scarce biomaterial, is essential for accurate analysis and proper interpretation of innovative results. This classification metric could enhance confidence in multiplexed proteomic platforms and improve the design of future panels.

## 1. Introduction

The growing interest in understanding the complex molecular mechanisms of different diseases and identifying novel biomarkers and potential drug targets has driven the development of highly multiplexed proteomic techniques. Moreover, the evolution of proteomic platforms has led to the evaluation of many analytes, enabling the simultaneous analysis of multiple samples and the development of different detection methods [1].

Mass spectrometry (MS) has been the gold standard technique in the field of proteomics, permitting the measurement of protein abundance, protein interactions, and posttranslational modifications, and providing crucial insights into multiple pathological mechanisms across various study areas [2,3]. Recently, two major affinity-based approaches capable of being used to analyze thousands of proteins have emerged as multiplex platforms [4]. Among these, the most well-known methods include immune-based techniques, such as Olink^®^ proteomics, and aptamer-based techniques, such as the SomaScan^®^ proteomic platform. While the Olink^®^ platform uses antibodies labeled with oligonucleotides to detect protein abundance by proximity extension assays (PEAs), quantitative polymerase chain reaction (qPCR), and next-generation sequencing (NGS), the SomaScan platform uses modified DNA aptamers that bind to proteins and detect them by fluorescence [5,6]. These high-throughput multiplex proteomic techniques represent valuable improvements by reducing the costs of single assays and reducing the time consumed by simultaneously analyzing multiple analytes and samples [7,8]. Furthermore, most of the proteomic analyses have been conducted using blood samples (plasma and serum), mainly because of the simple accessibility of that biomaterial. Thus, several studies have compared the affinity of these proteomic techniques (immune- and aptamer-based) for plasma samples in the context of several diseases, such as ovarian cancer, cardiovascular disease, atherosclerosis risk, and chronic obstructive pulmonary disease (COPD) [9,10,11,12,13]. In addition, other authors extended this analysis to cerebrospinal fluid (CSF) samples with a reduced sample size [14].

For neurodegenerative diseases, the use of high-throughput proteomic approaches could be of special interest since they could provide key information about the pathological changes occurring in the brain. Due to the inaccessibility of the brain, CSF, which is in direct contact with the central nervous system (CNS), is a well-established source of biomaterial reflecting brain protein alterations, among other outcomes. Specifically, in Alzheimer’s disease (AD), the main cause of dementia worldwide, a reduction in amyloid β 42 (Aβ42) levels and an increase in phospho-tau in threonine 181 (p-tau) in CSF have been widely described and used in memory clinics to aid in the diagnosis of these patients [15,16,17]. In this sense, proteomic profiling across the AD continuum could potentially provide insights into novel CSF biomarkers and pathways associated with disease development and would yield valuable information about AD pathological alterations [18,19,20,21].

However, due to the novelty of these techniques, the large number of proteins included, and the relevance of the potential findings, it is extremely important to perform a high-throughput assessment to determine the effects of biological and technical variability to ensure the reliability of the available platforms. Likewise, a validation analysis to improve the experimental design and reagents used could also strengthen the usage of these innovative technologies [22]. In this sense, few studies have rigorously assessed the impact of technical variations, preanalytical factors, and detection and quantification rates in CSF samples.

Consistent with this concept, we aimed to compare outcomes derived from the SomaScan^®^ 7k and Olink^®^ Explore platforms to investigate the reproducibility and reliability of these technologies. Our study included 1370 real-world CSF samples from the large Ace Alzheimer Center Barcelona (ACE) cohort composed of highly characterized subjects across the AD continuum [23]. Extensive characterization and comparison of these platforms were performed to critically elucidate their strengths and weaknesses and determine the gold standard of the thousands of proteins tested. Additionally, we also intended to identify the top-performing proteins for analyzing the associations between clinical phenotypes and AD core biomarkers in CSF.

## 2. Results

### 2.1. Proteomic Characterization and Demographics

In the subset of the ACE cohort, which included 264 individuals with available proteomic information derived from three independent experiments, there was a generally greater proportion of female individuals (mean age: 71.0 ± 8.28 years), most of whom were diagnosed with MCI. Additionally, those participants were highly characterized via biochemical analysis in CSF biofluid, including albumin, total globulins, total protein levels, the Qalb, and red blood cell count. In addition to those CSF biomarkers for AD, the ATN classification and the MMSE score of participants were also provided. The sample storage duration at −80 °C was also evaluated, and we did not observe any significant difference among controls and MCI individuals. Moreover, similar minimum (2.14 and 1.92 years) and maximum (5.33 and 5.85 years) sample storage durations were observed for controls and MCI individuals, respectively. Regarding the *APOE* locus, 29.5% of individuals were carriers of the ε4 allele, and lower MMSE scores were observed in individuals with MCI than in HCs, as we expected (Table 1).

### 2.2. PCA Evaluation

We conducted PCA to evaluate the variability captured by these proteomic platforms by aligning them with multiple orthogonal axes (PCs). In the nonadjusted correlation analysis between PCs and phenotypes, we found multiple statistically significant correlations with clinical traits across platforms (Figure 1c). However, the associations analysis, including all of these variables in the model, identified four independent components of the variance: CSF Aβ42, CSF p-tau, CSF total proteins, and the sample storage duration at −80 °C (Figure 1d). Few significant associations were found between the PCs and biochemical variables, such as plasma glucose levels and dyslipidemia, APOE genotype, disease status, and ATN classification. Interestingly, the first PC (PC1) of both SomaScanA and Olink Explore were strongly associated with CSF total protein, p-tau, and Aβ42 levels, suggesting that these variables are the major contributors to this fraction of explained variance. However, PC1 of the SomaScanB dataset was not significantly associated with any phenotype (Figure 1c,d). In addition, the correlation analysis of the top PCs revealed a strong correlation between SomaScanA and Olink Explore PC1 (r = 0.682, *p* < 2.2 × 10^−16^). In contrast, the SomaScanB PC1 exhibited a weaker negative correlation with the Olink Explore (r = −0.22, *p* = 8.73 × 10^−5^) and SomaScanA (r = −0.144, *p* = 0.0120) PC1. These results suggest that PC1 for both the SomaScanA and the Olink Explore datasets are tracking a similar source of variance and that there are other factors strongly impacting the SomaScanB variance (Appendix A). Other moderate correlations were found between Olink Explore PC1 and the second PC (PC2) for the SomaScan (SomaScanA: r = 0.532, *p* < 2.2 × 10^−16^; SomaScanB: r = 0.482, *p* < 2.2 × 10^−16^) as well as between SomaScanA PC1 and SomaScanB PC2 (r = 0.405, *p* = 1.81 × 10^−13^) (Appendix A).

Moreover, to assess the effect of demographic variables, such as age at LP and sex, on PC1 and PC2, we evaluated the PC plots according to these variables. For both the SomaScan^®^ and Olink^®^ Explore platforms, we observed a weak correlation between age and PCs (Appendix A). In the SomaScan^®^ platform, these PCs were not found to be associated with sex (Appendix A). Moreover, we found significant differences between males and females in terms of the Olink^®^ Explore PC1 and PC2 means (P_PC1_ = 4.9 × 10^−4^; P_PC2_ = 8.9 × 10^−8^; Appendix A).

Regarding the top protein loadings contributing to PC1 and PC2, we observed a reduced set of overlapping proteins in the two SomaScan^®^ experiments. This could be due to slight differences in the experimental analysis, mainly caused by noise or reagent saturation resulting in protein level variations. The proteins ADH1A (seq.17396.23), CD031 (seq.6604.59), FCRL6 (seq.6617.12), RB1 (seq.5024.67), and ULBP.1 (seq.3081.70) were represented in both of the SomaScan^®^ assays’ protein loadings with negative contributions to both PC1 and PC2 in the SomaScanA and SomaScanB datasets, respectively. Furthermore, no overlap was observed in the top 15 loadings between the SomaScan^®^ and Olink^®^ Explore platforms (Appendix A).

### 2.3. Comparing CVs

The CVs for both the SomaScan^®^ and Olink^®^ Explore platforms were calculated to analyze the precision of proteomic measurements using calibration samples [8]. In the intra-assay CV assessment (i.e., the variability of measures within a plate), several proteins with extreme CV values were identified via the Olink^®^ Explore platform (Appendix A). Moreover, excluding outliers at 1.5-fold IQR, Olink^®^ proteins that did not overlap with the SomaScan^®^ platform had more extreme CV values with different medians than did the overlapping proteins (*p* < 2.22 × 10^−16^). Additionally, we observed that Olink^®^ Explore intra-assay CV values had a more elongated distribution with different medians than those of the SomaScan^®^ CV values (*p* < 2.22 × 10^−16^), suggesting greater variability in Olink’s intra-plate precision. Additionally, no significant differences were detected in the median intra-assay CV when comparing overlapping and nonoverlapping SomaScan^®^ aptamers (Figure 2a, Appendix A).

Additionally, in the inter-assay CV assessment (i.e., variability between plates), the Olink^®^ Explore platform exhibited significantly more extreme CV values with different medians than those obtained with SomaScan^®^ (*p* < 2.22 × 10^−16^ for both overlapping and nonoverlapping proteins). Similarly, Olink^®^ proteins that did not overlap with the SomaScan^®^ platform showed significantly more extreme inter-assay CV values (*p* = 0.0011). In contrast, in the SomaScan^®^ platform, those proteins that did not overlap with Olink^®^ had less-extreme values compared with those that did overlap, and there was a statistically significant difference across these groups (*p* = 0.0043) (Figure 2b, Appendix A).

The median intra- and inter-assay CVs were greater for the Olink^®^ Explore platform than for the SomaScan^®^ platform (*p* < 2.22 × 10^−16^). Interestingly, we consistently observed that the median intra-assay CV was lower than the inter-assay CV for both proteomic techniques, suggesting that there is greater variability in calibration samples across multiple plates than in samples within the same plate (Figure 2, Appendix A).

To assess intra- and inter-assay CVs, we analyzed and compared CV quantiles across proteomic platforms. According to the intra-assay CV analysis, 98.6% and 68.9% of the proteins had CVs lower than 20% in the SomaScan^®^ and Olink^®^ Explore platforms, respectively. Conversely, inter-assay CV analysis revealed percentages of 93.7% and 30.0% for the respective platforms. Additionally, the median CV for the SomaScan^®^ platform reached approximately 5% for both intra- and inter-assay CVs. In contrast, the Olink^®^ Explore median CV exceeded 10% in both instances (Figure 2c). These findings indicate less variability in calibration samples within the same plate compared to measures analyzed across different plates, particularly with higher CV values observed in the Olink^®^ Explore panel.

### 2.4. Correlations Among Proteomic Measurements

#### 2.4.1. Intra-Platform Correlations in SomaScan^®^

A bimodal distribution with one mode at a rho of 0.1 with a reduced correlation was observed, as well as another mode with a high correlation at a rho of 0.85. This distribution suggests that a reduced number of aptamer measures are reproducible in different SomaScan^®^ CSF experiments (Figure 3a). The median Spearman rho values of the intra-platform correlation analysis was 0.302. Importantly, we identified a fraction of highly reproducible aptamers that were strongly correlated with a rho ≥ 0.5 (n = 2428, 33.3% corresponding to 2434 SomaScan pairs) (Figure 3a, Appendix A).

#### 2.4.2. Inter-Platform Correlations Between SomaScan^®^ and Olink^®^ Proteomics

When comparing the measurements of the same protein using the Olink^®^ and SomaScan^®^ techniques, we again observed a bimodal distribution in the inter-platform analysis. The majority of the CSF proteins exhibited low correlation coefficients, and a reduced portion of proteins were well reproducible in the other proteomic platform. There were 713 (26.7%) and 632 (23.7%) proteins with a rho ≥ 0.5 in the Spearman correlation with the SomaScanA and SomaScanB assays, respectively (Appendix A). The median Spearman rho values of the inter-platform correlation analysis were 0.097 and 0.092 for the SomaScanA and SomaScanB assays, respectively (Figure 3b,c). Additionally, we observed a similar distribution of the intra- and inter-platform Spearman rho values when stratifying by sex, suggesting no substantial differences between female and male individuals (Appendix A).

#### 2.4.3. Reproducibility and Reliability Metric

Considering the established correlation categories, we integrated these results into a single metric comprising nine categories for those proteomic measures that we assessed using both the SomaScan^®^ and Olink^®^ Explore platforms. We observed that from those 2428 highly reproducible aptamers with a Spearman rho ≥ 0.5 (corresponding to 2434 SomaScan–Olink pairs), only 676 proteins also correlated well with another proteomic technique (score 1). However, multiple proteins exhibited a lack of reproducibility in another proteomic platform (score 2: n = 171 and score 3: n = 376), which should be cautiously considered when conducting analyses (Table 2). Furthermore, we classified aptamer measures exclusively represented in SomaScan^®^ into categories (A–C). Among these, there were 1211 additional SOMAmers that were reproducible in two independent SomaScan^®^ assays with a Spearman rho ≥ 0.5 (Score A), which could be prioritized for further analysis in CSF (Table 2).

Based on the large number of analytes measured by the SomaScan^®^ platform and correlation analyses, we considered these SomaScan^®^ top-performing proteins with a good intra-platform correlation (score 1–3) and those not represented in the Olink^®^ Explore platform (score A) as candidates for subsequent analysis (n = 2428). These reliable measures might provide more accurate results in proteomic analysis.

### 2.5. PANTHER Annotations

For analyzing the protein classification with the PANTHER tool, we considered 6218 (97.1%) mapped proteins from 6402 unique UniProt IDs of SomaScan^®^ aptamer measures and 2872 (98.2%) unique mapped proteins from 2925 Olink^®^ Explore analytes. The top 20 PANTHER classifications indicated multiple significant annotations (FDR < 0.05) enriched in metabolite enzymatic conversion by transferases, hydrolases and oxidoreductases, protein modifying enzymes, and signaling processes (Appendix A). The molecular function and biological process annotations were significantly enriched in binding, catalytic activity (hydrolases and peptidases), signaling, response to stimulus, and regulatory mechanisms (Appendix A). In contrast, the cellular compartment annotations were significantly related to the cell surface, periphery, nucleus, organelle, and extracellular space (Appendix A). Interestingly, we observed several differences across platforms, and the Olink^®^ Explore platform was more enriched in immune proteins, including immunoglobulin receptor protein types, signaling, binding, and macromolecule metabolic processes. Additionally, these proteins were related to organelle cellular components. Conversely, the SomaScan^®^ platform was more enriched in signaling and protein-modifying enzyme protein classes and metabolic processes involving transferases and kinases. These proteins are commonly found in the cytoplasm, plasma membrane, and extracellular space. Both platforms had a similar proportion of proteins representing the top categories, except for the cellular compartment, where SomaScan^®^ proteins were more represented in the top categories (Appendix A).

Thereafter, considering reproducible SomaScan^®^ proteins (score 1–3 and A: n = 2428), we also explored the protein types and mechanisms involved using PANTHER. There were 2120 (97.4%) mapped proteins corresponding to 2177 unique UniProt IDs of SomaScan^®^ reproducible SOMAmers. The top-ranking significant protein classes (FDR < 0.05) were enriched in defense proteins, adhesion, and signaling protein classes compared to the complete set of aptamers represented in the SomaScan^®^ platform (Appendix A). For molecular function and biological process, reproducible proteins were related to signaling, metabolic regulation, and biogenesis (FDR < 0.05) (Appendix A). In addition, this set of proteins was enriched in cellular compartments located at organelle, plasma membrane, and nucleus locations compared to all SomaScan^®^ sets of proteins (Appendix A).

### 2.6. Linking SomaScan Platform Protein Signatures with CSF Biological Traits, Sample Demographics, and AD Endophenotypes

Interestingly, we observed a large proteomic signature of proteins that met the Bonferroni-corrected significance threshold (0.05/7289), with *p* values < 6.860 × 10^−6^ in the phenotype association analysis for age (n = 296 proteins), the Qalb (n = 462 proteins), CSF albumin (n = 420 proteins), CSF total globulin (n = 355 proteins), and CSF p-tau (n = 1175 proteins) phenotypes, considering age, sex, CSF total protein levels, and CSF biomarker technique (when applicable) as covariates. Other phenotypes showed a reduced proteomic signature with fewer associations, such as CSF red blood cell count (n = 107 proteins), CSF Aβ42 levels (n = 89 proteins), and sex (n = 30 proteins). Considering the Bonferroni-corrected threshold, we did not find any significant proteins associated with the MMSE score (Figure 4). However, a reduced subset of proteins was significant in five or more association analyses (n = 20) (Appendix A). The majority of significantly associated proteins were classified as reliable, suggesting that these associations were valid and had strong reliability (Figure 4, Appendix A).

Additionally, the variance explained in LASSO models by the reproducible set of proteins was greater than that explained by the complete set for age, CSF red blood cell count, CSF Aβ42, and p-tau phenotypes. However, we observed the opposite effect for CSF albumin, CSF total globulins, and the Qalb. None of the models had enough statistical power to explain the MMSE phenotype (Appendix A). Regarding sex, both models were highly similar, with an AUC of 0.9997 in the complete set (sensitivity = 0.984, specificity = 0.996) and an AUC of 0.9997 in the subset of good proteins (sensitivity = 0.986, specificity = 0.996), respectively (Appendix A).

Considering the complete set of aptamers analyzed in the SomaScan^®^ platform (n = 7289), we found 57 aptamers overlapping between the top 500 rankings of CSF Aβ42, p-tau, and MMSE, thus corresponding to 55 unique proteins (Figure 5a, Appendix A). Interestingly, we identified 88 aptamers corresponding to 83 unique proteins among reproducible proteins (good; n = 2428) in the SomaScan^®^ platform by intra-correlations, suggesting the presence of a general death signature in the CSF, as expected (Figure 5b, Appendix A). Notably, these differences were statistically significant, favoring the set of reproducible proteins, which exhibited a fivefold (504%) increase in overlapping hits among the selected AD endophenotypes compared to the full set of SomaScan^®^ aptamers (OR = 5.04, 95% CI [3.57–7.11], *p* = 2.11 × 10^−24^). This observation was further supported by empirical statistical methods. Specifically, the bootstrapping experiment revealed a mean simulated intersection of 2.113 proteins with a standard deviation of 1.441, which represents the expected result from random chance across 10,000 iterations. These results were significantly lower than the 57 aptamers identified as overlapping between the top-ranking CSF Aβ42, p-tau, and MMSE association analyses (P_e_ < 1 × 10^−4^) (Appendix A). Similarly, when we restricted the analysis to reproducible proteins with good intra-platform correlations (n = 2428), there were 88 overlapping aptamers between the top-ranking CSF Aβ42, p-tau, and MMSE score association analyses. This was also significantly greater than the mean simulated intersection of 15.6 proteins, with a standard deviation of 3.8 (P_e_ < 1 × 10^−4^) (Appendix A). As expected, the intersection between the top rankings of the three AD endophenotype associations was greater after only including reproducible protein measures than when considering the complete set of SomaScan^®^ proteins, most likely because the filtering of those reliable measures led to more accurate findings.

Additionally, we found that the most enriched mechanism of the top-ranking overlapping proteins in the complete SomaScan protein set was *Activation of BAD and translocation to mitochondria* (enrichment ratio = 148.068, FDR = 2.51 × 10^−8^) (Figure 5a, Appendix A). Similarly, considering the intersection of reproducible proteins associated with CSF AD biomarkers and MMSE score (Figure 5b), the most enriched mechanism was the *Activation of BAD and translocation to mitochondria* (enrichment ratio = 130.793, FDR = 5.86 × 10^−12^), as more overlapping genes were identified in the filtered analysis than in the complete set of SomaScan proteins.

The selection of reproducible proteins pointed to mechanisms such as *Kinase maturation complex 1* (enrichment ratio = 76.637, *p* = 1.23 × 10^−5^) and *Neurexins and neuroligins* (enrichment ratio = 26.275, *p* = 1.85 × 10^−4^), which improved their performance compared to that of the complete set. Additionally, three mechanisms increased the representation of the reproducible proteins compared to the complete set: the *EGF receptor signaling pathway* (enrichment ratio = 17.060, FDR = 5.59 × 10^−5^), *Transcriptional regulation by TP53* (enrichment ratio = 8.063, FDR = 6.11 × 10^−5^), and *Regulation of protein localization to membrane* (enrichment ratio = 12.470, FDR = 8.45 × 10^−5^) (Appendix A). Additionally, nonintersecting proteins were evaluated through enrichment analysis, revealing similar mechanisms represented by the CSF p-tau and MMSE rankings considering the complete set of SomaScan proteins (n = 7289) and the reproducible proteins (good; n = 2428) (Appendix A).

## 3. Discussion

The use of high-throughput proteomic platforms has enabled the simultaneous evaluation of multiple analytes [24]. Considering the novelty of these techniques and the many proteins assessed, it is extremely important to perform comprehensive quality control to ensure the reliability of these measures. In this sense, comparing the performance of the SomaScan^®^ and Olink^®^ Explore platforms could provide essential information for understanding the strengths and limitations of these techniques by assessing the reproducibility and the impact of preanalytical factors and technical variations. To our knowledge, here, we present the largest head-to-head comparison of CSF sample analysis using two of the main multiplex proteomics platforms available to date, the SomaScan^®^ and Olink^®^ panels. Several authors have compared different proteomic platforms, mainly for their use in analyzing plasma or serum biomarkers. Thus, our analyses provide the novelty of using the ACE cohort, which includes CSF samples obtained from extensively characterized real-world HC and MCI patients from a memory clinic. This valuable dataset and the reported results are notably relevant for providing information about brain pathological changes occurring in a wide variety of diseases, including AD.

Our analysis showed that SomaScan^®^ measurements were more uniform in both intra- and inter-assay CV evaluations compared to the Olink^®^ Explore panels. Although there is a reduced number of published studies reporting CV values in CSF SomaScan^®^ data, our results were consistent with previous findings in plasma, in which values near 5% were reported by Gold et al. [6] and other authors [8,25,26,27,28]. However, the number of sample controls or calibrators could contribute to an increase in the variability of CV observations. The Olink^®^ Explore panels included only two pooled plasma samples in a plate, which is considerably lower than the five samples included in the SomaScan platform.

In our PCA, we observed that PC1 in both SomaScan^®^ and Olink^®^ explained a similar proportion of the total variance in both datasets, suggesting that both of these high-throughput proteomic platforms are capable of capturing a similar fraction of explained variance, and tracking a similar source of variance. Plasmatic biochemical variables show a limited association with the proteomic principal components on both platforms, suggesting a limited translatability between of results between biofluids. These results also suggested that the SomaScanB PC1 dataset might be influenced by alternative factors beyond our analysis capacity compared to other datasets (SomaScanA or Olink Explore). Despite including a different number of proteins in these platforms, they might explain a similar proportion of the underlying biological mechanisms. Previous studies have suggested that a high proportion of CSF proteomic measures are highly correlated with each other and form clusters, supporting the idea that PC1 could explain a large proportion of variance [29]. Additionally, we found a similar general representation of PANTHER categories, suggesting that these proteins are involved in similar molecular mechanisms and pathways despite a few differences that are exclusive to each platform. Moreover, we assessed which proteins were the major contributors to PC1 and PC2. Interestingly, we observed a reduced number of proteins represented in both the SomaScanA and SomaScanB top rankings of protein loadings, suggesting that this approach is sensitive to extreme proteomic measures and preanalytical factors that could affect the protein ordering and composition of each PC. The top-ranking Olink^®^ Explore protein loadings for PC1 and PC2 included several proteins that were mainly studied in the inflammation panel.

The bimodal distribution found in both intra- and inter-correlation analyses suggests that a fraction of these aptamers did not perform proper dimensional recognition of proteins, thus making it difficult to compare multiple measurements between the two proteomic techniques. Similarly, Dammer et al. also compared SomaScan^®^ 5k with the Olink platform and MS using CSF samples. Although they used a reduced sample size (n = 36), they reported a high median rho parameter (rho ~ 0.7) [14]. They enriched the analysis of well-correlated aptamers by selecting pairs of reagents with the same UniProt ID and the best correlation, which might have led to an overestimation of the rho parameters. In a later study, Dammer et al. increased their sample size (n = 300) and observed a median rho parameter of 0.59, suggesting that the reduced sample size and aptamer selection impacted their results [30]. Nevertheless, the presence of protein quantitative trait loci (pQTLs), posttranslational modifications, and alternative splicing variants could explain why these subsets of nonreproducible aptamer measures (moderate or low correlation coefficient) created alternative proteoforms that were differentially detected in each platform. This poor correlation could also be caused by technical factors, including preanalytical factors such as sample handling and off-target binding, due to a potential lack of specificity and nonspecific noise behind these protein measurements. However, Hok-A-Hin et al. assessed the effect of several preanalytical factors in the CSF proteome and reported that the majority of proteins measured in the SomaScan^®^ and Olink^®^ platforms remained stable after these extreme conditions [31].

Other studies using plasma proteomic data pointed to these factors impacting the performance of high-throughput analysis [10,32,33]. A wide variety of studies have described several poorly correlated plasma analytes while analyzing the correlation of aptamer-based (SomaScan^®^ 1.1k, 1.3k, and 7k) and antibody-based techniques (Olink^®^, Myriad-RBM multiplex panel, and Mesoscale Discovery Platform) [10,33]. Moreover, several plasma proteomic studies have reported a bimodal distribution in the correlation between various SomaScan^®^ (5k and 1.3k) and Olink^®^ (Explore and 92-protein) panels, suggesting that one or both platforms were potentially affected by these confounding factors [8,32,34]. These studies conducted in plasma reported results in line with our findings and support the notion that bimodal distributions are related to common technical issues unrelated to any specific properties of the biofluid under study. Further studies are needed to validate these results, especially at the protein level, which would be useful for evaluating the reliability of each high-throughput proteomic measurement.

Furthermore, we established a metric accounting for reproducibility and reliability. Here, we will introduce a resource focused on the reproducibility and translatability of protein markers in CSF. It will be available as a downloadable resource with unrestricted access for the scientific community. Notably, a wide variety of these proteins classified in the top 20 rankings of the reproducibility score have been studied in the context of AD pathological mechanisms or endophenotypes. These include immunoglobulin M [35], aggrecan [36], chitotriosidase-1 [37], haptoglobin [38], IL1RL1 [39], CD177 [40], CRP [41], SIRBP1 [42], VNN2 [43], and carbonic anhydrases [44]. The developed classification metric will provide valuable insights for future CSF proteomic analyses using these techniques, as well as ensuring the validity of the obtained results. Further research is needed to evaluate the correlation between CSF and plasma proteomic results, which will be highly valuable for exploring potential translatability of this metric across these biofluids.

Moreover, we found that the aptamers most strongly associated with clinical phenotypes and AD core biomarkers were classified into the “Good” or reproducible proteomic category, suggesting that these results are genuine and reliable. Several of these significant associations have already been described in the literature, such as associations of CSF neurofilament proteins (NFL and NFH) with age [45,46] and myosin light chain 4 (MYL4) with sex [47], among others [48,49,50,51,52,53,54,55,56,57]. Interestingly, we observed that a large proportion of CSF albumin and Qalb associations, classified as good proteins, were negative, while the opposite effect was observed in the CSF total globulin associations where good proteins had positive associations. These findings might suggest that CSF albumin levels might interfere with the protein binding of SomaScan^®^ aptamers. It is tempting to speculate that multiple conformational changes in the proteome are mediated by albumin and affect protein detection because of the general scavenging properties of albumin, which might involve interactions with numerous aptamers. Therefore, further studies are needed to validate our hypothesis and elucidate the potential role of albumin in aptamer binding using high-throughput proteomic platforms. The combination of this process, proteome variability, and other preanalytical factors might significantly impact SomaScan^®^ measures, leading to a weaker correlation. Additionally, the use of different strategies for protein recognition and quantification methods (fluorescence in SomaScan^®^ and qPCR/NGS in Olink^®^ Explore) could potentially lead to differences in detection and reproducibility across these platforms. Additionally, we analyzed the overlap between the top-ranking proteins associated with the MMSE score and the core AD biomarkers: CSF Aβ42 and p-tau. We observed a significant fivefold increase in the number of overlapping proteins with the selection of good proteins, which is much greater than what was expected from bootstrapping analyses. These results strongly support the validity of our strategy for selecting well-reproduced proteins for further analysis, highlighting the importance of extensive QC, and providing valuable insights into the reproducibility of proteomic platforms. This information can enhance the validation and interpretation of proteomic results as well as contributing to the standardization proteomic assays, leading to an improved understanding AD physiopathology and diagnosis.

Our analysis also had several limitations. First, the lack of publicly available technical details of these proteomic assays hampered the comprehensive evaluation of additional factors potentially impacting the assay performance. Second, both proteomic techniques used different detection methods; SomaScan^®^ fluorescence intensity was measured via microarray hybridization, and Olink^®^ Explore used qPCR and NGS, which could have led to differences in affinity between platforms measuring multiple proteoforms. Third, we calculated CVs using raw data before QC to mitigate bias from following the consensus recommendations of each technique in data curation. Thus, the presence of outlier proteomic measures or low-quality reads might influence the CV distribution. Fourth, the preselection and enrichment of specific molecular routes and proteins performed by both platforms might have an impact on PANTHER annotations, which could in turn alter the identified subset of reliable and reproducible proteins. Finally, multiple proteins were not represented in the Olink^®^ Explore panels and could not be validated using another platform. Alternative validation methods are needed to evaluate these proteins exclusively represented in one proteomic platform, such as exploring alternative proteomic methods, such as Elisa or MS, targeting the protein of interest. Further studies using several larger proteomic platforms are needed to assess this subset of proteins.

## 4. Methods and Materials

### 4.1. Study Participants and Selection Criteria

A total of 1370 CSF samples were provided by the ACE CSF cohort, which was composed of healthy control (HC) participants and individuals diagnosed with mild cognitive impairment (MCI) or dementia. Briefly, syndromic diagnosis was established at the Memory Clinic of ACE (Barcelona, Spain) by a multidisciplinary group of neurologists, neuropsychologists, and social workers. Healthy controls and individuals with subjective cognitive decline (SCD) who showed no objective evidence of cognitive impairment in the evaluation and a Clinical Dementia Rating (CDR) of 0 [58] were classified as HCs. A diagnosis of MCI was given to patients with one or more impaired cognitive domains on the neuropsychological battery of ACE (NBACE) [59], accounting for the cut-offs of impairment for age, formal education levels, and a CDR of 0.5 [59,60,61,62,63]. The 2011 National Institute on Aging and Alzheimer’s Association (NIA-AA) guidelines were used to establish the AD dementia diagnosis [15]. Further information on the clinical characteristics of these individuals has been described elsewhere [23,64,65].

The lumbar puncture (LP) for the assessment of CSF AD-related biomarkers was offered to (a) patients with MCI and dementia assessed at ACE’s memory clinic [64]; (b) participants of the Fundació ACE Healthy Brain Initiative (FACEHBI) [66] with SCD; and (c) participants in the BIOFACE study with early-onset MCI [67,68]. We collected a CSF sample from the LP following the consensus recommendations [69], centrifuged it (2000× *g* for 10 min at 4 °C), aliquoted it, and stored it at −80 °C. For CSF Aβ42, total tau (t-tau) and p-tau protein determination, an aliquot was defrosted on the day of the analysis and vortexed at room temperature. These protein levels were examined using a standard enzyme-linked immunosorbent assay (ELISA) kit (Innotest β-AMYLOID (1-42), Fujirebio Europe, Göteborg, Sweden) or the Lumipulse G600II automated platform (Fujirebio Inc.) [64]. Additionally, all CSF samples underwent complementary biochemical analysis.

### 4.2. SomaScan Proteomic Profiling

SomaScan^®^ 7k proteomic profiling (SomaLogic Operating Co., Inc., Boulder, CO, USA) was selected as the representative aptamer-based detection technology. This multiplexed proteomic technique involves the use of 50 µL of CSF per sample and modified DNA aptamers, also called *SOMAmers,* to measure 7596 proteins. These *SOMAmers* bind to protein targets, and the abundance of these complexes was detected by using fluorescence in a conventional DNA array [6]. The protein amount was expressed in relative fluorescent units (RFUs) and normalized using the adaptive normalization by maximum likelihood (ANML) method [70]. Two aliquots from each subject were run in two batches using the same SomaScan^®^ platform (SomaScan 7k, version 4.1) and analyzed within 6 months. For these proteomic analyses, we used multiple aliquots in the first thawing cycle. The first experiment included 632 samples (SomaScanA) selected considering the ATN (amyloid, tau, neurodegeneration) status [15] and diagnosis of dementia with CDR > 0.5 or nondementia with CDR ≤ 0.5. The second experiment comprised 1370 samples (SomaScanB) included randomly, with 46.1% of individuals overlapping between the two SomaScan^®^ assays (Appendix A). An additional quality control step involved a log2 transformation to adjust to a normal distribution, and z scores were calculated using the R function *scale* with centering and scaling, applied separately to each analysis.

### 4.3. Olink Proteomic Profiling

Olink^®^ proteomics (Uppsala, Sweden) was selected as the representative antibody-based detection technology. This platform uses antibodies labeled with oligonucleotides to detect protein amounts by PEA, a combination of qPCR and NGS technologies [5]. However, Olink^®^ proteomic profiling was performed on a single batch of 510 samples using the complete Olink^®^ Explore panel, which measures a total of 2944 proteins (Cardiometabolic, Cardiometabolic II, Inflammation, Inflammation II, Neurology, Neurology II, Oncology, Oncology II in November 2021) (Uppsala, Sweden). The results are expressed as log2 normalized protein expression (NPX) values. Additionally, we scaled the protein measures using the R function *scale* with centering and scaling. Overall, 22.3% and 37.1% of individuals overlapped with the SomaScanA and SomaScanB assays, respectively (Appendix A).

For platform comparison purposes, we selected a subset of 305 samples with available proteomic data from three assays (Olink^®^ Explore, SomaScanA, and SomaScanB). These proteomic platforms included 7289 human aptamers from SomaScan^®^ and 2943 analytes from the Olink^®^ Explore panels that passed quality control within each technique. In addition, we found 2161 overlapping reagents between platforms, corresponding to 2159 unique proteins according to UniProt ID, representing 26.8% of the pairs assessed (Appendix A). We based our analysis on the previously published proteomic head-to-head comparison of Katz et al. [8].

### 4.4. Protein Annotation Using PANTHER

We used the PANTHER classification system version v17.0 [71] to elucidate the subfamilies and functions of the measured proteins and annotate them by using a *Homo sapiens* reference gene list. A statistical overrepresentation test was performed to associate these proteins with PANTHER GO-Slim terms such as Cellular Compartment, Biological Process, and Molecular Function using Fisher’s exact test with FDR correction in the PANTHER online tool. In addition, functional classification was performed to annotate the protein class of each analyte.

### 4.5. Principal Component Analysis (PCA)

We performed PCA on log2-transformed scaled data from both proteomic platforms using the package *tidymodels* in R 4.1.1. We evaluated the percentage of variance explained by each principal component (PC), the number of PCs needed to explain 95% of the variance, and the 15 proteins with the greatest absolute contributions to the first and second PCs. The SomaScan^®^ platform needed fewer PCs to explain 95% of the variance (N_PCs SomaScanA_ = 218, N_PCs SomaScanB_ = 197) than the Olink^®^ Explore platform (N_PCs_ = 258) (Figure 1a). Similarly, the variance explained by the first PC was slightly greater for the SomaScan^®^ platform (SomaScanA = 20.7%, SomaScanB = 18.6%) than for the Olink^®^ Explore panels (18.43%) (Figure 1b). Additionally, we conducted Pearson correlations and linear associations between the PCs and multiple potentially noisy clinical variables and phenotypes to evaluate their contributions to the variance explained. Additionally, we conducted Pearson correlations between the top 5 PCs to assess whether they represented a similar fraction of explained variance.

Finally, using the interquartile range (IQR) method, we identified and removed outlier individuals outside 1.5-fold of the IQR. We excluded 21 and 19 outlier individuals from the SomaScanA and SomaScanB datasets, respectively, from subsequent analysis. Additionally, we excluded 7 outlier individuals from the Olink^®^ Explore dataset for further analysis. These outlier individuals did not have gross differences in clinical characteristics across all SomaScan^®^ assays, and slight differences were found across proteomic studies in CSF biomarkers due to sample selection (Appendix A). Hence, 264 individuals were considered for subsequent analysis.

### 4.6. Coefficient of Variation (CV)

Similar to Katz et al., we used two pooled calibration samples to calculate CVs in the Olink^®^ Explore data [8]. Concerning the SomaScan^®^ data (SomaScanB analysis), we selected the first 2 calibration samples of each plate to consider the same number of calibrators in both proteomic platforms. Intra-assay CVs were computed for the 2 QC samples on each plate and averaged across all plates. Interassay CVs were calculated using 76 pooled samples in 39 plates from Olink^®^ and 32 samples in 16 plates from SomaScan^®^. Additionally, we calculated the 10th, 25th, 75th, and 90th CV percentiles and the median to evaluate the differences between both proteomic platforms, as described elsewhere [8]. To calculate the CVs, we used the following equations:(1)$${CV}_{SomaScan}=\sigma /\mu$$
(2)$${CV}_{Olink\;Explore}=\sqrt{e^{{\left(ln(2)\times \sigma_{NPX}\right)}^{2}}-1}$$

### 4.7. Pairing Platform Reagents by Protein Target

We matched protein measurements from the SomaScan^®^ and Olink^®^ platforms using the UniProt identification code (UniProt ID), which links each assay with a peptide. Several SomaScan^®^
*SOMAmers* measuring the same protein were identified, and in the same way, several proteins were independently measured in different Olink^®^ Explore subpanels. Furthermore, protein complexes identified with multiple UniProt IDs were considered a match in both platforms if there was a full correspondence between the complete set of identification codes.

### 4.8. Correlation of the Matched Reagents

To analyze the intra-platform correlation between the two SomaScan^®^ assays, we performed a Spearman correlation analysis on overlapping individuals. We classified protein measures according to their Spearman rho coefficient: (1) good (rho ≥ 0.5), (2) moderate (0.3 ≤ rho > 0.5), and poor (rho < 0.3). Similarly, an additional analysis was performed to evaluate the inter-platform correlation between protein measurements derived from both SomaScanA and SomaScanB and Olink^®^ Explore assays. Replication of proteins in both platforms was also classified into the same categories described above.

Finally, to integrate the intra- and inter-assay correlation analyses into a single metric accounting for reproducibility and reliability, we aggregated the established correlation categories (Table 3). For each intra-platform category (good, moderate, and poor), we assessed the corresponding classification for inter-platform correlations, considering both available SomaScanA and SomaScanB assays. We assigned the highest reproducibility and reliability to a good intra-platform correlation with a rho ≥ 0.5 and established an additional ordering considering the inter-platform correlation classification (score 1–3). If a given protein had different inter-platform classification categories for the SomaScanA and SomaScanB experiments, the category with the highest rho value was considered for the score. The same process was followed for the other intra-platform categories until a single metric was established (Table 3). Thereafter, we also established an additional classification for those SomaScan^®^ measures that were not represented in the Olink^®^ Explore panels based on internal reproducibility (intra-platform correlation) in the SomaScan^®^ platform (Table 3).

### 4.9. Associations Between Clinical Traits and CSF Biomarkers

Linear regressions were performed to analyze associations between scaled log2-transformed protein levels measured using the SomaScan^®^ platform and (1) clinical traits such as age, sex, CSF albumin, CSF total globulins, CSF red blood cell count, the albumin quotient (Qalb; a measure of blood–brain barrier (BBB) leakage), and the Mini-Mental State Examination (MMSE) score closest in time to the LP; and (2) CSF biomarkers such as CSF Aβ42 and p-tau, all of which were collected from the ACE cohort. To avoid redundancy, we did not analyze the association with t-tau levels, as it is widely known that both protein levels are highly correlated [16,72]. We considered sex, age, CSF total protein levels, and CSF biomarker technique (when applicable) as covariates. We also performed logistic regressions to analyze protein associations with sex. Additionally, we considered the reliability and reproducibility of SomaScan proteins while analyzing each phenotype association. Then, we compared the top 500 proteins, ordered by significance, associated with CSF Aβ42, CSF p-tau, and MMSE score to evaluate the overlap and characterize these overlapping proteins using the WebGestalt tool [72] considering the *genome protein-coding* reference gene list. We conducted a chi-square test to evaluate differences in overlapping proteins between the reproducible and complete sets of SomaScan proteins.

To assess variance explained by the phenotypes of interest and redundancy in the protein dataset caused by highly correlated aptamer measures, we performed LASSO regression using (1) the complete set of SomaScan^®^ proteins (n = 7289) and (2) reproducible proteins with rho ≥ 0.5 in the intra-assay correlation (good, n = 2428). The LASSO model is a 5-fold cross-validation method that is repeated 5 times using the *Caret* R package version 6.0-94. A total of 80% of the samples were randomly selected to train the model, and the remaining 20% were selected for testing purposes. Several lambda values were analyzed to select the optimal regularization parameter. Moreover, we adopted the *glmnet* method, which uses the *ROC* metric for categorical variables and the *RMSE* for continuous numerical variables.

Thereafter, we also performed a bootstrapping experiment to analyze the overlap between three sets of 500 randomly selected proteins (with replacement), and this process was repeated for 10,000 iterations. The aim was to compare the simulated number of overlapping proteins with the actual results obtained from the top-ranking overlap of proteins significantly associated with CSF biomarkers and MMSE score and to assess potential differences in the expected overlap from random chance (simulated) and our actual results. We considered two different datasets with (1) 7289 proteins (complete SomaScan^®^ set) and (2) 2428 SomaScan^®^ proteins with rho ≥ 0.5 in the intra-platform correlation analysis.

## 5. Conclusions

Our results demonstrate that our strategy effectively evaluated the reproducibility and reliability of SomaScan^®^ proteomics in CSF samples from a real-world cohort, providing valuable insights into protein validation and developing a metric to classify reliable proteins. Furthermore, this strategy also provides crucial information about the comparability with antibody-based proteomic platforms, which could help to validate and interpret the results obtained in other studies. Further research is needed to extensively characterize and compare these platforms to determine their strengths and weaknesses and to determine the gold standard for analyzing thousands of proteins.

## Figures and Tables

**Figure 1 ijms-26-00286-f001:**
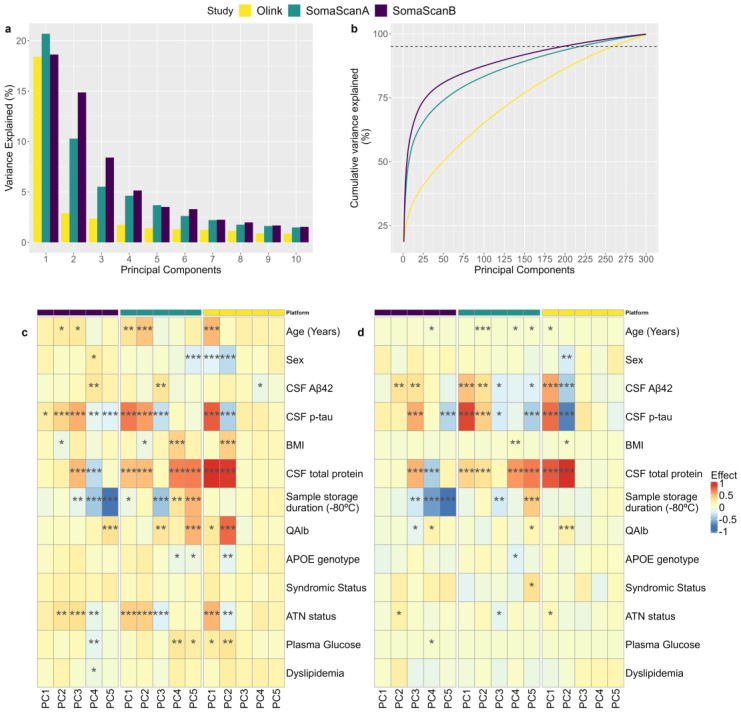
Principal component analysis results. (**a**) Variance explained by the top 10 PCs. (**b**) Percentage of cumulative variance explained by the top 300 PCs; the dashed line represents 95% of the explained variance. (**c**) Pearson correlations of the top 5 PCs with clinical phenotypes according to the nonadjusted model. (**d**) Linear model associations of the top 5 PCs adjusted by all clinical phenotypes that were included in the model. An asterisk (*) represents a *p* value lower than 0.05, two asterisks (**) represent a *p* value lower than 0.01, and three asterisks (***) represent a *p* value lower than 0.001.

**Figure 2 ijms-26-00286-f002:**
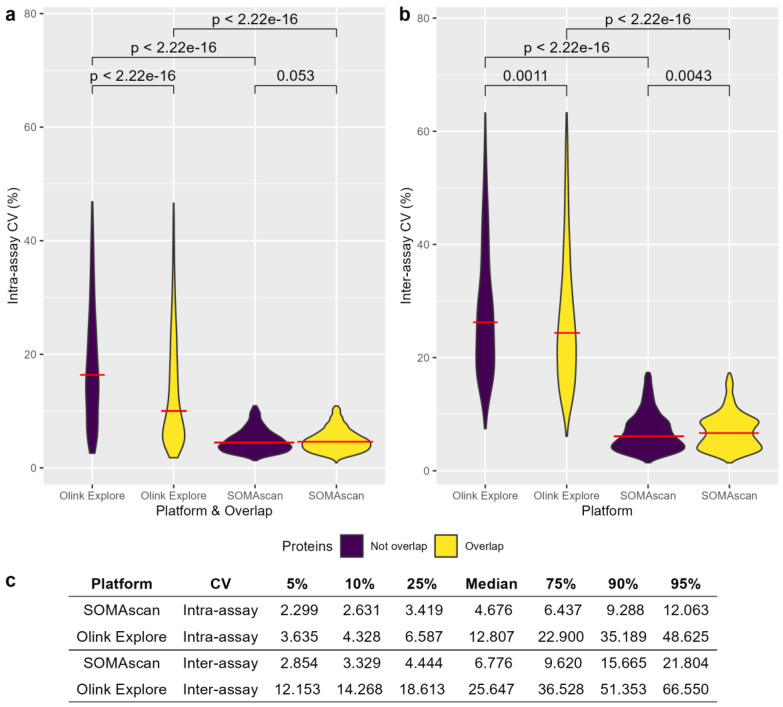
Coefficient of variation (CV) for the SomaScan^®^ and Olink^®^ Explore Platforms. (**a**) Intra- and (**b**) inter-assay CVs colored by the overlap between the SomaScan^®^ and Olink^®^ assays after removing outlier proteins at 1.5-fold IQRs. The Mann-Whitney-Wilcoxon test was applied to explore differences across median values. The red line in the zoom plot represents the median CV for each platform. (**c**) Percentages of intra- and inter-assay CVs for the SomaScan and Olink Explore platforms.

**Figure 3 ijms-26-00286-f003:**
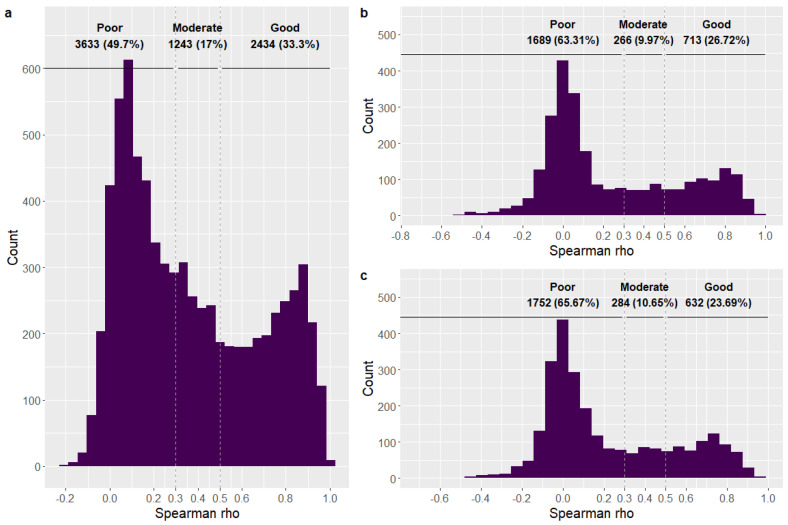
Distribution of Spearman’s rho values in the correlation analysis, including the sample size of each category (n (%)). (**a**) Intra-platform correlation between SomaScanA and SomaScanB assays. (**b**) Inter-platform correlation between SomaScanA and Olink Explore. (**c**) Inter-platform correlation between SomaScanB and Olink Explore platforms. We established three categories: good (rho > 0.5), moderate (0.5 > rho ≥ 0.3), and poor (rho < 0.3).

**Figure 4 ijms-26-00286-f004:**
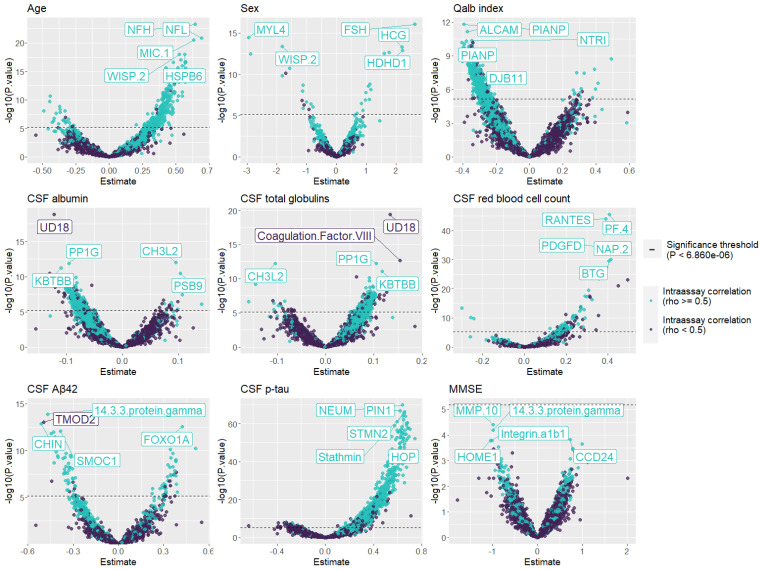
Volcano plots of the associations between SomaScan^®^ proteins and clinical phenotypes. Proteins with intra-assay correlations (rho ≥ 0.5) are colored light blue, and the dashed line represents the significance threshold (*p* value < 6.860 × 10^−6^).

**Figure 5 ijms-26-00286-f005:**
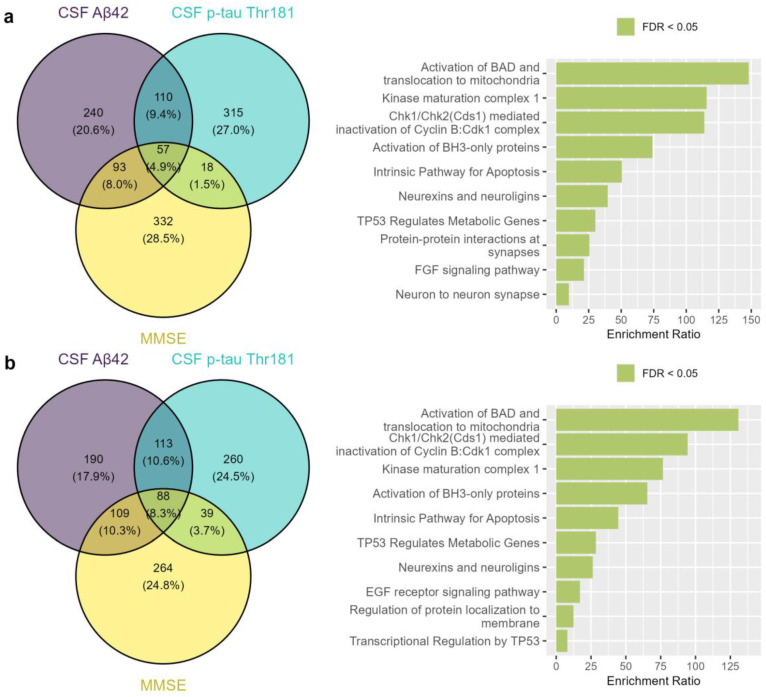
Venn diagram of CSF Aβ42, CSF p-tau, and MMSE associations and enrichment analysis using the WebGestalt tool. (**a**) Complete set of SomaScan proteins (n = 7289), including 57 overlapping aptamers between the top 500 rankings. (**b**) Reproducible SomaScan proteins with good intra-correlations (n = 2428), including 88 overlapping aptamers between the top 500 rankings.

**Table 1 ijms-26-00286-t001:** Demographic and clinical data of the ACE cohort (n = 264).

	Control	Mild Cognitive Impairment	Overall	*p* Value
		(MCI)		
	(n = 33)	(n = 230)	(n = 264)	
**Sex (female)**	18 (54.5%)	129 (56.1%)	147 (55.7%)	0.7
**Mean age (SD), years**	64.9 (5.54)	71.9 (8.25)	71.0 (8.28)	**<0.0001**
**CSF biomarkers**				
** * CLEIA Lumipulse * **	3 (1.1%)	71 (27%)	74 (28.1%)	
Mean Aβ42 (SD)	1210 (121)	977 (393)	987 (388)	0.31
Mean p-tau (SD)	33.7 (4.73)	54.2 (31.8)	53.3 (31.5)	0.26
** * ELISA * **	30 (11.4%)	159 (60.5%)	190 (71.9%)	
Mean Aβ42 (SD)	1080 (202)	760 (323)	810 (328)	**<0.0001**
Mean p-tau (SD)	43.3 (9.97)	70.2 (36.6)	65.7 (35.2)	**0.0004**
**ATN (%)**				**<0.0001**
A-T-N-	33 (100%)	120 (52.2%)	154 (58.3%)	
A+T-N-	0 (0%)	6 (2.6%)	6 (2.3%)	
A+T+N+	0 (0%)	104 (45.2%)	104 (39.4%)	
**Mean Qalb (SD)**	841 (115)	865 (98.0)	862 (100)	0.31
**Mean CSF albumin (SD)**	0.236 (0.107)	0.240 (0.088)	0.239 (0.0902)	0.96
**Mean CSF total globulins (SD)**	0.200 (0.0676)	0.190 (0.061)	0.191 (0.0616)	0.54
**Mean CSF total protein (SD)**	0.459 (0.173)	0.452 (0.141)	0.453 (0.145)	0.94
**Mean CSF red blood cell count (SD)**	47.2 (71.1)	54.6 (142)	53.9 (135)	0.9
**Mean MMSE score (SD)**	29.6 (0.565)	25.5 (3.44)	26.0 (3.49)	**<0.0001**
Missing values n (%)	6 (18.2%)	4 (1.7%)	11 (4.2%)	
***APOE* e4 carriers (%)**	5 (15.2%)	73 (31.7%)	78 (29.5%)	0.092
**Mean sample storage duration (−80 °C), years**	4.58 (0.731)	4.26 (0.768)	4.30 (0.768)	0.084

**Note**: CSF: cerebrospinal fluid; CLEIA: chemiluminescent enzyme-immunoassay (Lumipulse automated platform); ELISA: enzyme-linked immunosorbent assay (Innotest, Lumipulse). Significant *p* values were highlighted in bold. The *p* value was calculated using the Fisher’s exact test for proportions and Wilcoxon rank sum test (*getDescriptionStatsBy* R function).

**Table 2 ijms-26-00286-t002:** Combination of intra- and inter-platform correlation categories for creating a single reproducibility and reliability score.

Intra-Platform Correlation (SomaScan vs. SomaScan)	Inter-Platform Correlation (SomaScan vs. Olink Explore)	Score	n (%)
GOOD	GOOD	1	676 (9.248)
	MODERATE	2	171 (2.339)
	POOR	3	376 (5.144)
MODERATE	GOOD	4	45 (0.616)
	MODERATE	5	71 (0.971)
	POOR	6	311 (4.254)
POOR	GOOD	7	5 (0.068)
	MODERATE	8	32 (0.438)
	POOR	9	981 (13.42)
GOOD	Not available Olink	A	1211 (16.566)
MODERATE	Not available Olink	B	816 (11.163)
POOR	Not available Olink	C	2615 (35.773)

**Note**: The score of 1–9 comprised values for proteins represented in both SomaScan^®^ and Olink^®^ Explore proteomic platforms (n = 2668), and A–C for nonoverlapping proteins between platforms (n = 4642).

**Table 3 ijms-26-00286-t003:** Intra- and inter-platform correlation categories.

Intra-Platform Correlation (SomaScan vs. SomaScan)		Inter-Platform Correlation (SomaScan vs. Olink Explore)		Score
GOOD	rho ≥ 0.5	GOOD	rho ≥ 0.5	1
		MODERATE	0.5 > rho ≥ 0.3	2
		POOR	rho < 0.3	3
MODERATE	0.5 > rho ≥ 0.3	GOOD	rho ≥ 0.5	4
		MODERATE	0.5 > rho ≥ 0.3	5
		POOR	rho < 0.3	6
POOR	rho < 0.3	GOOD	rho ≥ 0.5	7
		MODERATE	0.5 > rho ≥ 0.3	8
		POOR	rho < 0.3	9
GOOD	rho ≥ 0.5	-	-	A
MODERATE	0.5 > rho ≥ 0.3	-	-	B
POOR	rho < 0.3	-	-	C

**Note**: rho refers to the spearman correlation coefficient. Proteins not represented in both proteomic platforms were only classified according the intra-platform correlation in three additional categories (A–C).

## Data Availability

The data that support the findings of this study are publicly available from the corresponding authors upon reasonable request. Additionally, the largest preprocessed SomaScan^®^ proteomic dataset, also known as SomaScanB, has been uploaded and is publicly accessible through the Alzheimer’s Disease Data Initiative (ADDI) community.

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
