# Peer review of "Head-to-Head Comparison of Aptamer- and Antibody-Based Proteomic Platforms in Human Cerebrospinal Fluid Samples from a Real-World Memory Clinic Cohort"

_ijms, 2024, doi:10.3390/ijms26010286_

Round 1
Reviewer 1 Report
Comments and Suggestions for Authors
The authors present an interesting paper in which they examine the performance of two reputable platforms utilised in current day biomarker discovery; namely SomaScan and Olink. In utilising CSF samples acquired from patient populations that had attended Alzheimer clinics, the original sample set was reduced significantly in order to eliminate any variables such as BMI, sample storage duration, etc. that may have impacted on the hypothesis in question. With the reduced sample size (>200), reproducibility and correlations between the two approaches was examined, with significant overlap determined between the two platforms. Taken together, these types of studies are important in determining the validity of data obtained from clinical samples given the scarcity of such in certain contexts, and the approach taken by the authors is detailed and informative.
In reviewing the manuscript I made a couple of observations. The following should be considered by the authors when preparing a suitable revision.
1. Information on the age of the samples employed (in other words, how old was the oldest sample employed and how old was the youngest sample – not how old were the participants who donated the samples) and the means of storage, frequency of thawing etc. would be interesting in order to determine potential variation across the sample set employed. While this was examined somewhat, it would be useful if textually this information was included, and the criteria of the sample set employed.
2. The formatting of the data is good for the most part, however, the labelling/text in many instances is too light/small to be readily readable. In particular, Figure 4 requires some work to make the results more legible. The authors should consider improving this aspect in any resubmission.
Author Response
Comment 1: Information on the age of the samples employed (in other words, how old was the oldest sample employed and how old was the youngest sample – not how old were the participants who donated the samples) and the means of storage, frequency of thawing etc. would be interesting in order to determine potential variation across the sample set employed. While this was examined somewhat, it would be useful if textually this information was included, and the criteria of the sample set employed.
Response 1: We agree with this comment and therefore we have added the duration of the sample storage and the number of freeze-thaw cycles information into the demographics table (Table2, Page 8, line 323), as well as a brief comment in the Method (Page 3, line 152) and Results section (Page 8, line 316). Additionally, the SomaScanA samples were selected as part of another project based on the diagnosis of dementia with a CDR>0.5, non-dementia diagnosis with CDR≤0.5 and ATN status. Conversely, the SomaScanB samples were included randomly. We have added the selection criteria to the Methods section (Page 4, line 153-155).
Comment 2: The formatting of the data is good for the most part, however, the labelling/text in many instances is too light/small to be readily readable. In particular, Figure 4 requires some work to make the results more legible. The authors should consider improving this aspect in any resubmission.
Response 2: Thank you for pointing this out, we have revised multiple figures to ensure that the labels are legible.
Reviewer 2 Report
Comments and Suggestions for Authors
The manuscript ijms-3312862 entitled Head-to-head comparison of aptamer- and antibody-based proteomic platforms in human cerebrospinal fluid samples from a real-world memory clinic cohort by Raquel Puerta and coworkers evaluated the reproducibility and reliability of aptamer-based (SomaScan® 7k) and antibody-based (Olink® Explore 3k) proteomic platforms in cerebrospinal fluid (CSF) samples from the Ace Alzheimer Center Barcelona real-world cohort. The 12-category metric of reproducibility combining both correlation analyses identified 2,428 highly reproducible SomaScan CSF measures, with over 600 proteins well reproduced on another proteomic platform. The association analyses among AD clinical phenotypes revealed that the significant associations mainly involved reproducible proteins. The validation of reproducibility in these novel proteomics platforms, measured using this scarce biomaterial, is essential for accurate analysis and proper interpretation of innovative results.
The work is scientifically sounding and well written.
Results are interesting.
Tables and figures are informative.
Discussion is consistent with results.
The references are appropriated.
English is very good.
Line 260: table 1 should stay in one page
Line 407: the notes to figure3 should stay in one page.
Author Response
Comment 1: The manuscript ijms-3312862 entitled Head-to-head comparison of aptamer- and antibody-based proteomic platforms in human cerebrospinal fluid samples from a real-world memory clinic cohort by Raquel Puerta and coworkers evaluated the reproducibility and reliability of aptamer-based (SomaScan® 7k) and antibody-based (Olink® Explore 3k) proteomic platforms in cerebrospinal fluid (CSF) samples from the Ace Alzheimer Center Barcelona real-world cohort. The 12-category metric of reproducibility combining both correlation analyses identified 2,428 highly reproducible SomaScan CSF measures, with over 600 proteins well reproduced on another proteomic platform. The association analyses among AD clinical phenotypes revealed that the significant associations mainly involved reproducible proteins. The validation of reproducibility in these novel proteomics platforms, measured using this scarce biomaterial, is essential for accurate analysis and proper interpretation of innovative results.
The work is scientifically sounding and well written.
Results are interesting.
Tables and figures are informative.
Discussion is consistent with results.
The references are appropriated.
English is very good.
Line 260: table 1 should stay in one page
Line 407: the notes to figure3 should stay in one page.
Response 1: We sincerely appreciate your comments, and we have checked that tables and figure captions are completely placed in one page.
Reviewer 3 Report
Comments and Suggestions for Authors
This fascinating and relevant study expands the knowledge regarding the biomarkers for AD diagnosis and monitoring. The authors performed robust techniques and presented their data with quality and clarity. I must congratulate the researchers for their significant contributions. Overall, no flaws were detected, but I identified a few aspects that could be better addressed, as the following:
1) Has any study conducted a similar investigation to other neurodegenerative diseases or related ones?
2) Can the CSF parameters be correlated with any other plasma parameter to define a pattern? CSF is a critical sample and challenging to obtain. If it was possible to establish a correlation with plasma parameters, it would be relevant. Do the authors have any insight about it? It would be interesting to approach briefly such a topic in the discussion.
3) PCA evaluation is a valuable tool for enhancing the generally limited comprehension of crude data. In the authors' opinion, is it possible to correlate or evaluate ordinary lab data (glucose and lipid levels) with the proteomic evaluation to track a potential biomarker fingerprint of AD?
4) The study primarily evaluated samples from female patients. Does this provide insight into potential differences in the obtained findings compared to male patients? Please, better discuss such aspect.
Author Response
Comment 1: 1) Has any study conducted a similar investigation to other neurodegenerative diseases or related ones?
Response 1: Thank you for your comment. We have evaluated similar studies in the Introduction (Page 2, line 83-87) and Discussion section (Page17, line 611-639). Briefly, the majority of analyses comparing SomaScan and Olink Explore have been conducted in plasma or serum biomaterials in the context of several diseases such as COPD, cancer or cardiovascular disease. Dammer et al. performed a similar approach comparing the cerebrospinal fluid (CSF) measures of these platforms. However, our analysis includes a larger sample size and we consider all proteins included in both SomaScan and Olink Explore platforms without making any preselection of proteins potentially leading to bias.
Dammer, E. B. et al. Multi-platform proteomic analysis of Alzheimer’s disease cerebrospinal fluid and plasma reveals network biomarkers associated with proteostasis and the matrisome. Alzheimer’s Research & Therapy 2022 14:1 14, 1–32 (2022).
Comment 2: 2) Can the CSF parameters be correlated with any other plasma parameter to define a pattern? CSF is a critical sample and challenging to obtain. If it was possible to establish a correlation with plasma parameters, it would be relevant. Do the authors have any insight about it? It would be interesting to approach briefly such a topic in the discussion.
Response 2: We appreciate your valuable comment. We agree that the CSF is challenging to obtain and plasma measures could be also relevant to evaluate. In our experience, we have analysed the correlation between plasma and CSF proteomic measures in a reduced subset of immune-related proteins using the SomaScan technology. We have observed a reduced correlation between inflammatory mediators in plasma and CSF, suggesting that there might be different inflammatory processes ongoing in both compartments. Moreover, there is scarce information about plasma and CSF correlations for other proteins, and specially for the subset of reliable proteins in which we are focused. We plan to expand our results on this matter in future studies using our plasma data. Lastly, we have added a comment to our manuscript empathising the need for further studies to elucidate the potential correlation between plasma and CSF proteomic analytes extending to the complete proteome (Page 18, line 650).
Morató X. et al. Associations of plasma SMOC1 and soluble IL6RA levels with the progression from mild cognitive impairment to dementia. Brain, Behavior, & Immunity – Health 2024 42 (2024)
Comment 3: 3) PCA evaluation is a valuable tool for enhancing the generally limited comprehension of crude data. In the authors' opinion, is it possible to correlate or evaluate ordinary lab data (glucose and lipid levels) with the proteomic evaluation to track a potential biomarker fingerprint of AD?
Response 3: We agree with the reviewer that the PCA is useful to enhance the comprehension of large datasets. Following this idea, we have evaluated the correlation of proteomic principal components, serum glucose and alterations in lipid levels (dyslipidemia) biochemical alterations. We did not observe a clear association between these parameters suggesting that the proteomic levels might not be influenced by these biochemical variables (Figure 1). We have expanded this information in the manuscript in the Results section (Page 9, line 331) and the discussion section (Page 16, line 590).
Comment 4: 4) The study primarily evaluated samples from female patients. Does this provide insight into potential differences in the obtained findings compared to male patients? Please, better discuss such aspect.
Response 4: Thank you for your comment. Our real-world cohort includes a higher proportion of female individuals, comprising a 55.7% overall, but this extra 6% of females is not significantly contributing to sex-related differences. This increased proportion of female individuals is consistent with the higher prevalence of AD and other dementia in females (Alzheimer’s Association Facts and Figures 2024). We have conducted a correlation analysis stratifying by sex, and we observed a very similar distribution of Spearman rho in both sexes. We have added these results to the Supplementary Figure 9 and expanded this information on the Results section (Page 12, line 424). Additionally, we want to highlight that the sex was used as a covariate in further associations and lasso models to minimize potential effects of this variable in the results.
Alzheimer’s Association. 2024 Alzheimer’s Disease Facts and Figures. Alzheimers Dement 2024;20(5). Link: https://www.alz.org/media/Documents/alzheimers-facts-and-figures.pdf

Reviewer 4 Report
Comments and Suggestions for Authors
The study titled "Head-to-Head Comparison of Aptamer- and Antibody-Based 2 Proteomic Platforms in Human Cerebrospinal Fluid Samples from a Real-World Memory Clinic Cohort" compares two advanced proteomics platforms, SomaScan® and Olink®, for analyzing cerebrospinal fluid (CSF) proteins, focusing on their reproducibility and limitations. It highlights differences in protein detection and suggests ways to improve biomarker validation for diseases like Alzheimer’s.
However, I have a few comments and Recommendations that could enhance the manuscript:
Recommendations
- Provide More Technical Insights
Please offer a deeper comparison between the detection methods of SomaScan® (fluorescence) and Olink® (qPCR/NGS) and discuss how the differences in these techniques might affect the sensitivity, specificity, and reproducibility of CSF proteomics. - Address Limitations Proactively
Please suggest ways to address key limitations, such as the lack of overlapping proteins for validation. For example:
- Propose alternative validation methods for proteins unique to each platform.
- Explore the potential for developing hybrid platforms or new approaches that combine the strengths of both technologies.
- Discuss the impact of calculating coefficient of variations using raw data before quality control and how this could influence the interpretation of variability metrics.
- Expand on Novel Hypotheses with Evidence
Further develop speculative hypotheses, such as the influence of albumin on aptamer binding. It is better to add an additional section with theoretical evidence to back these ideas. - Clarify Clinical Implications
Please provide more specific guidance on how the study’s results can influence clinical practice, especially for Alzheimer’s disease biomarker validation and diagnosis.
- Please suggest clear next steps for applying the reproducibility metric in proteomics research and its potential role in standardizing assays.
- The conclusion section is missing.
The conclusion section is currently missing. Please add a concise conclusion emphasizing the translational potential of your findings, focusing on how they can promote adoption in both clinical practice and research settings.
Author Response
Comment 1: Provide More Technical Insights: Please offer a deeper comparison between the detection methods of SomaScan® (fluorescence) and Olink® (qPCR/NGS) and discuss how the differences in these techniques might affect the sensitivity, specificity, and reproducibility of CSF proteomics.
Response 1: Thank you for your valuable comments and suggestions. We have detailed the technical aspects of our analyses in the Method section and further elaborated on them in the Discussion section. Additionally, we have improved our manuscript by expanding the Discussion section regarding the technical insights of these platforms (Page 18, line 668).
Comment 2: Address Limitations Proactively: Please suggest ways to address key limitations, such as the lack of overlapping proteins for validation. For example:
- Propose alternative validation methods for proteins unique to each platform.
- Explore the potential for developing hybrid platforms or new approaches that combine the strengths of both technologies.
- Discuss the impact of calculating coefficient of variations using raw data before quality control and how this could influence the interpretation of variability metrics.
Response 2: Considering your recommendation, we have improved the limitations section by adding alternative validation methods and improving the explanation of coefficient of variation calculation using raw data (Page 18, line 690,694).
Comment 3: Expand on Novel Hypotheses with Evidence: Further develop speculative hypotheses, such as the influence of albumin on aptamer binding. It is better to add an additional section with theoretical evidence to back these ideas.
Response 3: Regarding the albumin hypothesis, this hypothesis was developed based our results, but we did not find theoretical evidence to support it. Additionally, we have commented in the Discussion section that further studies are needed to validate our hypothesis and elucidate the potential role of albumin in protein detection using high-throughput proteomic platforms (Page 18, line 665). Thank you for pointing this out.
Comment 4: Clarify Clinical Implications: Please provide more specific guidance on how the study’s results can influence clinical practice, especially for Alzheimer’s disease biomarker validation and diagnosis. Please suggest clear next steps for applying the reproducibility metric in proteomics research and its potential role in standardizing assays.
Response 4: Thank you for your recommendation, we have expanded the Discussion section to further explain the clinical implications of our reproducibility analyses, their impact in the understanding of AD pathology and future research (Page 18, line 677).
Comment 5: The conclusion section is missing: The conclusion section is currently missing. Please add a concise conclusion emphasizing the translational potential of your findings, focusing on how they can promote adoption in both clinical practice and research settings.
Response 5: Thank you for your comment, we have added a conclusion section (Page 18, line 700).
Reviewer 5 Report
Comments and Suggestions for Authors
Make corrections as attached

need corrections
Author Response
Comment 1: The reviewer provided language revisions in the pdf document of the manuscript.
Response 1: Thank you for your language revision. The quality of English language has been improved in the following sections, and our collaborator Dr. Marchant, who is native English speaker, has reviewed the language in the manuscript.
- Page 1, line 44
- Page 2, line 68
- Page 2, line 75
- Page 2, line 89
- Page 3, line 111
- Page 3, line 121
- Page 3, line 156
- Page 4, line 162
- Page 4, line 169
- Page 5, line 220
- Page 6, line 241
- Page 7, line 274
- Page 7, line 287
- Page 12, line 452
- Page 18, line 649
- Page 18, line 687
- Page 19, line 713